# Examining implicit procedural learning in tetraplegia using an oculomotor serial reaction time task

Ayala Bloch[1,2]*, Michal Shaham[1], Eli Vakil[3,4], Simone Schwizer Ashkenazi[3,4], Gabi Zeilig[5,6]

1 Department of Behavioral Sciences, Ariel University, Ariel, Israel, 2 The National Institute for the Rehabilitation of the Brain Injured, Tel Aviv, Israel, 3 Leslie and Susan Gonda (Goldschmied) Multidisciplinary Brain Research Center, Bar-Ilan University, Ramat Gan, Israel, 4 Department of Psychology, Bar-Ilan University, Ramat Gan, Israel, 5 Neurological Rehabilitation Department, Sheba Medical Center, Ramat Gan, Israel, 6 Sackler Faculty of Medicine, Tel Aviv University, Tel Aviv, Israel

* ayalabl@shikumil.org.il

## Abstract

### Background and objective

Clinical observations indicate that implicit procedural learning, a central component of physical and psychosocial rehabilitation, is impeded following spinal cord injury. In accordance, previous research has revealed a specific deficit in implicit sequence learning among individuals with paraplegia using a standard, manual version of the serial reaction time task. To extend these findings and shed light on the underlying sources of potential spinal cord injury-related deficits in sequence learning, we used an ocular activated serial reaction time task to compare sequence learning performance between individuals with tetraplegia and healthy controls.

### Participants and measures

Twelve participants with spinal cord injury in C5-T1 were compared to 12 matched control participants on measures derived from an ocular activated serial reaction time task. Depression and additional cognitive measures were assessed to explore the source and specificity of potential sequence learning deficits.

### Results

Like controls, and in contrast with previous findings in paraplegia, the spinal cord injury group showed intact implicit sequence learning, evidenced by declining reaction times and improved anticipation over the first six blocks of the serial reaction time task, and an advantage for the initial learning sequence over a novel interference sequence.

### Conclusions

The ocular activated serial reaction time task elicited a performance pattern similar to standard motor versions, such that participants with tetraplegia demonstrated unimpaired

**Data Availability Statement:** All relevant data are within the manuscript and its Supporting Information files.

**Funding:** The author(s) received no specific funding for this work.

**Competing interests:** The authors have declared that no competing interests exist.

sequence learning. This suggests that previously reported implicit sequence learning deficits in spinal cord injury directly involved motor functioning rather than cognitive aspects of the task, and that the ocular activated sequence learning task could be a valid alternative for assessing implicit sequence learning in populations that cannot perform spinal-cord dependent motor tasks. Implications for post-spinal cord injury rehabilitation and adjustment are discussed.

## Introduction

Implicit procedural learning, or the development of routine skills without reliance on conscious or explicit memory processes [1], can have far-reaching effects on the rehabilitation and overall well-being of individuals with spinal cord injury (SCI). This is because new skills inevitably determine the ability to effectively and independently conduct daily functions following SCI, such as communicating using eye or head motions, or proficiently activating a wheelchair using limbs with newly limited motion. Clinically observed difficulties in skill-acquisition and routine-learning following SCI have led researchers to examine these abilities in the lab.

When SCI does not preclude motor function of the upper limbs, implicit procedural learning can be examined with manual sequence learning tasks. Bloch et al. [2] examined individuals with paraplegia using the serial reaction time (SRT) paradigm, a sequence learning task commonly used to study learning- and memory-related behaviors including implicit learning of a motor skill [3, 4]. In the standard version, speed of manual responses to stimuli presented in a repeated sequence is compared to response times for stimuli presented in a novel sequence. When procedural memory is intact, reaction times decrease gradually during the original sequence blocks, presumably indicating implicit learning, and then increase sharply during the novel, unlearned block. This pattern is absent or less pronounced in populations with damage to the basal ganglia [5, 6], a neural area associated with procedural learning.

In accordance with clinical observations in paraplegia, Bloch and colleagues [2] reported a more moderate learning curve on the SRT as compared to controls, and no increase in reaction time with the introduction of a novel sequence, suggesting that this population did not learn the initial sequence. This difference could not be explained by other study variables, including mood, intelligence, and verbal and visuospatial memory, supporting the possibility that difficulties in skill-acquisition and routine-learning experienced by patients during post-SCI rehabilitation reflect specific implicit procedural learning deficits associated with their injuries. In explaining their findings, the researchers noted that implicit procedural learning deficits could be related to secondary changes in the brain caused by reduction of afferent signals from affected limbs. Indeed, a broad range of studies has documented post-SCI reorganization, inflammation, and degeneration in sensorimotor and additional brain areas [7, 8], among them the basal ganglia-thalamocortical pathways believed to play a primary role in implicit procedural learning [9].

A logical progression from this line of work was to examine procedural learning in individuals with SCI resulting in partial or total loss of use of all four limbs (tetraplegia). This posed a greater challenge, as a manual task could not be employed. However, a recent study on healthy participants suggested that an ocular activated version of the SRT (O-SRT) task could be a viable alternative for assessing implicit sequence learning in populations that can move their eyes but are unable to perform motor tasks engaging the spinal cord [10]. Eye movement-based responses in individuals with tetraplegia have been used similarly in other tasks requiring

verbal or manual responses [11, 12]. Directly comparing an ocular activated version of the SRT task to the manual version described above, Vakil and colleagues demonstrated that the extent of sequence learning measured by the two versions was essentially identical in healthy participants. Furthermore, by tracking eye movements in response to a blank slide appearing before each target, they evaluated a novel anticipation measure believed to be a purer indication of sequence learning.

In the current study, we used this ocular activated task to compare sequence learning performance between individuals with tetraplegia and healthy controls, to extend the findings of Bloch et al. [2] and to shed light on the underlying sources of potential SCI-related deficits in sequence learning. The decreased sequence learning found using the manual version of the SRT in the paraplegia study [2] could stem from deficits in spinal cord-dependent motor components of implicit learning or from other (non-motor) cognitive processes underlying this skill. Oculomotor responses, however, are not spinal cord-dependent, as they bypass peripheral nerve involvement. Thus, sequence learning deficits, if found, would reflect deficits in cognitive rather than spinal cord-dependent motor components of implicit learning. In contrast, if the reduced sequence learning observed in paraplegia was dependent on response modality and specific to the manual motor response, then the oculomotor responses required by the O-SRT task would not be affected, and participants in the tetraplegia and control groups could be expected to perform the task similarly.

## Methods

### Participants

Nineteen individuals with SCI resulting in tetraplegia were initially recruited, during or after their rehabilitation at the Chaim Sheba Medical Center Department of Neurological Rehabilitation. All had acquired C5-T1 SCI, graded as American Spinal Injury Association (ASIA) Impairment Scale (AIS) A or B [13]. Three measures were employed to indicate the absence of concomitant brain injury among the participants: 1. absence of post-traumatic amnesia (PTA) [14]; 2. Glasgow Coma Scale (GCS) [15] rating above 13 (14/15 or 15/15); and 3. absence of neuroimaging findings indicating brain injury, when available. Additional exclusion criteria included: impaired vision, learning disability, below average performance on verbal and performance IQ measures (see Measures section below), history of alcohol or drug abuse, premorbid psychiatric diagnoses, and depression scores above the mild range, as assessed by the Quick Inventory of Depressive Symptomatology Self-Report (QIDS-SR) [16]. Based on these criteria, seven participants were excluded. The remaining 12 participants (one female) had a mean age of 43.3 years (range: 25–62; standard deviation: 14.2) and a mean time from injury of 3.75 years (range: 0.5–7; standard deviation: 2.2). Patients received various medications in accordance with their personal treatment plans, previous to and during participation in the study. Clinical and demographic information for the experimental group is detailed in Table 1.

Twelve control participants were included in the study, after being recruited through social networks and personal acquaintance with the researchers and screened to rule out the exclusion criteria noted above. To decrease variability, they were each matched to an SCI group participant with respect to age, sex, and education (see Table 2). According to a dependent samples $t$-test, their mean age of 43.3 years (range: 24–62; standard deviation: 14.1) did not differ significantly from that of the SCI group ($p = 1$). To minimize potential confounding variables and improve matching, we also assessed between-group differences in verbal and performance IQ (all tools described in the Measures section below) and found no between-group differences in any of the measures ($p < .05$).

Table 1. Spinal cord injury group: Demographic and clinical information.

| Participant | Sex | Age (years) | Education | Cause | AIS | LOI | Years since injury |
|---|---|---|---|---|---|---|---|
| 1 | M | 25 | High School | ST | B | C6 | 4 |
| 2 | M | 52 | University | MVA | A | C7 | 7 |
| 3 | M | 31 | University | MVA | A | C4 | 6 |
| 4 | M | 39 | High School | ST | A | C5 | 4 |
| 5 | M | 62 | High School | ST | A | C5 | 6 |
| 6 | M | 57 | University | Cervical myopathy | B | C4 | 5 |
| 7 | M | 54 | University | ST | A | C4 | 3 |
| 8 | F | 48 | High School | MVA | A | C5 | 0.5 |
| 9 | M | 61 | High School | MVA | A | C5 | 1 |
| 10 | M | 26 | University | MVA | A | C5 | 2.5 |
| 11 | M | 25 | High School | ST | B | C5 | 1 |
| 12 | M | 39 | University | ST | B | C5 | 5 |

Abbreviations: AIS, American Spinal Injury Association Impairment Scale; LOI, level of injury as assessed by the International Standards for Neurological Classification of Spinal Cord Injury (ISNCSCI); MVA, trauma in motor vehicle accident; ST, trauma during sport or recreation activity. High school = graduated from high school; University = undergraduate degree at least.

The study was approved by the Chaim Sheba Medical Center ethics committee. All participants entered voluntarily. When physically able, they signed a written informed consent form; otherwise, fingerprints were used to indicate informed consent.

## Measures

**International Standards for Neurological Classification of Spinal Cord Injury (ISNCSCI).** Severity of SCI was based on neurological level of injury (NLI; defined by lowest motor and sensory intact segment) and completeness or incompleteness of neurological damage, as defined by AIS grades A and B. AIS A indicates complete injury, with no sensory or motor function preserved in sacral segments S4-S5. AIS B indicates preservation of sensory but not motor function below the 'zone of injury' and includes sacral segments.

Table 2. Control group: Demographic information.

| Participant | Sex | Age (years) | Education |
|---|---|---|---|
| 1 | M | 24 | High School |
| 2 | M | 51 | University |
| 3 | M | 31 | University |
| 4 | M | 38 | High School |
| 5 | M | 60 | High School |
| 6 | M | 62 | University |
| 7 | M | 53 | University |
| 8 | F | 47 | High School |
| 9 | M | 60 | High School |
| 10 | M | 27 | University |
| 11 | M | 26 | High School |
| 12 | M | 40 | University |

High school = graduated from high school; University = undergraduate degree at least.

**Ocular activated SRT (O-SRT) task.** In the current study, the O-SRT paradigm introduced by Vakil et al. [10] was employed. The task was programmed in E-Prime 2.0 and eye movements were recorded using the SMI (SensoMotoric Instruments, Teltow, Germany) iView 120 REDm Eye Tracker.

Stimuli included five slides, each with a resolution of 1400 × 1050 pixels, with four white squares arranged in a diamond shape on a grey background. A black dot (indicating the target) appeared in one of the four white squares. The size of each square was 6 × 6 cm and the diameter of the dot was 1.5 × 1.5 cm (see Fig 1), based on the layout described in Kinder et al. [17]. Four slides included a target image, and the fifth slide did not display a target (this slide is referred to as the blank slide used to measure anticipation). The stimuli were presented on an LCD computer screen (size 42 × 24 cm; resolution 1600 × 900 pixels). The recording device was installed beneath the screen. Participants were seated in front of the screen, approximately 60 cm away from it.

In each trial, participants were instructed to find the target and to look at it until it disappeared. The slide was activated by oculomotor responses, such that it was presented until the participant fixated on the square that contained the target either for 100 ms, or for 1000 ms if the participant did not fixate on the target for the required duration. The experiment consisted of eight blocks with 1-minute intervals between the blocks. Each experimental block consisted of a 12-element sequence repeated 9 times. Thus, each block was composed of 108 trials. There were 6 learning blocks (Block 1 to Block 6), an interference block with a different sequence (Block 7), and an additional block with the original sequence (Block 8). Each block began the sequence from a different point. The sequences were adapted from Gabriel et al. [18] and no first-order predictive information was provided (i.e., each location was preceded by the same location only once—12, 13, 14, 21, 23, etc.). Both contained one reversal (Sequence 1: 1–2–1; Sequence 2: 3–2–3). The order of the sequences was counterbalanced such that for half of the participants the learning sequence was 3–4–2–3–1–2–1–4–3–2–4–1, and the interference sequence was 3–4–1–2–4–3–1–4–2–1–3–2. For the other half, the order was reversed. Each number in the sequence was matched with one of the four squares: 1, 2, 3, and 4 to correspond with down, left, right, and up, respectively. Calibration was conducted at the beginning of the experiment using a standard 5-point grid for both eyes. A 4-point grid was used for validation

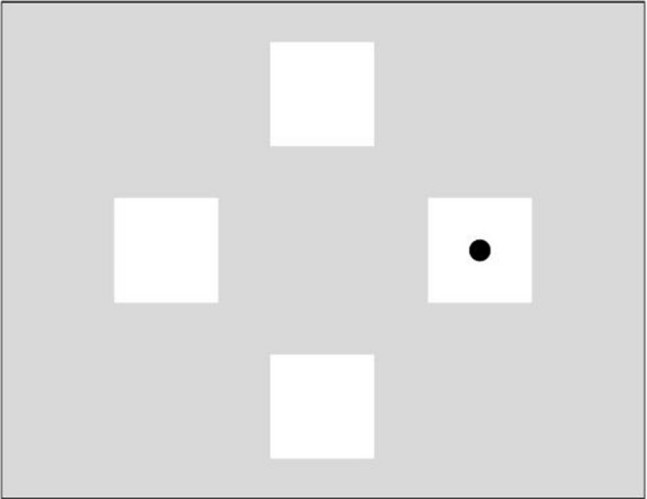

**Fig 1. Example of ocular activated serial reaction time task target slide.**

after each calibration trial. When accuracy was derived in more than 0.8˚, calibration and validation were repeated.

The measures derived from the task included the changes in reaction time (in response to the target slide) and percentage of correct anticipations for the following intervals: *learning* (Block 1 to Block 6); *interference* (Block 6 to Block 7); and *recovery* (Block 7 to Block 8).

**Awareness.** At the end of Block 8, participants were first asked two questions to assess awareness of the repeated order. The first question was, 'Did you notice anything special about the experiment?' (One point was given for a 'yes.') It was followed by the more specific question, 'Did you notice any patterns during the experiment?' (One point was given for indication that there was a repeated sequence.) The measure derived from this part of the task, *Awareness*, had a maximum value of 2.

**Depression.** The Quick Inventory of Depressive Symptomatology Self-Report (QIDS-SR) [14] was employed to assess depression according to DSM-IV criteria. The questionnaire has been used previously to assess depressive symptoms following SCI [19, 20] and its sensitivity is equal to self-report depression measures typically used in the past [21]. It contains 16 multiple choice questions with four answers each (0–3). Scores range from 0 (no depression) to 27 (major depression). To compare means accurately between the groups, we divided total scores by the maximal score of 27.

**Additional cognitive measures.** The Vocabulary, Matrix Reasoning, and Digit Span subtests of the Hebrew version (first edition) of the Wechsler Adult Intelligence Scale (WAIS-III) [22] were also administered.

## Procedure

Tests, tasks, and questionnaires were administered in one or two (up to one month apart) sessions, in the following order: O-SRT task, awareness questions, QIDS-SR, WAIS-III subtests.

## Data analysis

All statistics analyses were performed using IBM SPSS Statistics Professional 20.0, with a 0.05 level of significance.

O-SRT data were registered using BeGaze™ (SensoMotoric Instruments, Teltow, Germany). As in Vakil et al. [10], there were two dependent measures: speed (reaction time for the slide with the target) and percentage of correct anticipations (for the anticipation slide). Three phases of performance were analyzed separately: learning (Blocks 1 to 6), interference (Block 6 vs. Block 7), and recovery from interference (Block 7 vs. Block 8).

Reaction time was calculated based on *entry time*, or the time interval preceding the first fixation on the square in which the target appeared. The mean of the per-block median reaction times for every 12-item sequence (9 medians per 108-trial block) was analyzed. Three two-way matched-subject repeated measures analyses of variance (ANOVAs) were performed with the between-subject variable Group (SCI versus control) and the within-subject variable Block, to assess the change in reaction time in the three measures derived from the task, as follows: learning (reaction time change in blocks 1–6), interference (reaction time change following series change, blocks 6–7), and recovery (reaction time change following return to learned sequence, blocks 7–8).

Anticipation score per block was evaluated based on the transition of gaze to the correct subsequent position during presentation of the blank slide that followed each target slide. We used the 'area of interest' function in the BeGaze program and enlarged the squares into a triangle, so that four triangles covered the four squares and the center point of the screen. During the 500 ms in which the blank slide was presented, gaze (as measured by the location of the

fixations) could: (1) remain in the location where the previous target had appeared, (2) move to more than one location, including or not including the correct location of the subsequent target (the final location determined whether response was considered correct or incorrect), (3) move to only one of the incorrect locations, or (4) move to only the correct location. For each 12-item sequence, we calculated the percentage of correct anticipations by dividing the number of fixations on the correct location (option 4) by the total number of fixations on a single location (options 3+4). We then calculated the mean for nine sequences per block (similar to the way reaction time was calculated) to establish the percentage of correct anticipations for each block for all participants. As with reaction time, two-tailed paired sample $t$-tests were then used to compare the percentage of correct anticipations of the control and experimental groups on the first block of the task, followed by three two-way matched-subject design Group by Block RM ANOVAs for the learning, interference, and recovery phases. A related samples Wilcoxon signed-rank test was used to compare the control and SCI groups with respect to the Awareness measure.

Related samples Wilcoxon rank-signed tests were used to compare the control and SCI groups with respect to scores derived from the WAIS-3 subtests.

## Results

### O-SRT task

**Reaction time.** The results of the O-SRT task reaction time analyses (described above) are presented in Fig 2.

*Learning.* Across groups, there was a significant reduction in RT over Blocks 1–6, $F(5, 55) = 15.41$, $p < .001$, $\eta_p^2 = .58$, while neither the main effect of Group, $F(1, 11) = 0.904$, $p = .36$, $\eta_p^2 = .08$, nor the Group x Block interaction, $F(5, 55) = 0.27$, $p = .93$, $\eta_p^2 = .02$, was significant. These results indicate that both groups improved in performance over blocks and showed similar learning patterns.

*Interference.* There was a main effect of Block, $F(1, 11) = 25.24$, $p < .001$, $\eta_p^2 = .70$, indicating that RTs in interference Block 7 were significantly higher than in the preceding Block 6 (i.e., interference effect), across groups. Neither the main effect of Group, $F(1, 11) = 0.86$, $p = .38$, $\eta_p^2 = .07$, nor the Group x Block interaction, $F(1, 11) = ,07$, $p = .8$, $\eta_p^2 = .006$) was significant, indicating that the two groups showed similar reaction times across blocks in this stage, and similar interference patterns.

*Recovery.* There was a main effect of Block, $F(1, 11) = 18.70$, $p < .001$, $\eta_p^2 = .63$, indicating lower reaction times in Block 8 than in Block 7 (i.e., recovery effect), across groups. Neither

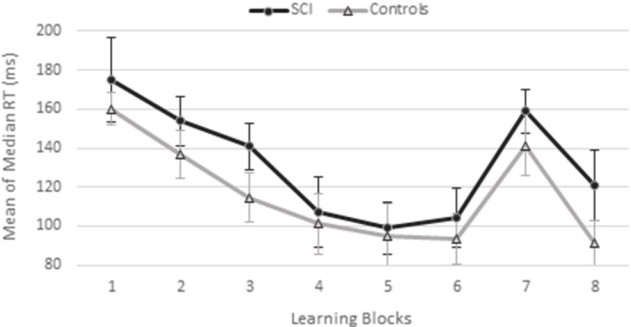

**Fig 2. Ocular activated serial reaction time task reaction times (RT; mean and SEM) in the spinal cord injury (SCI) and control groups.**

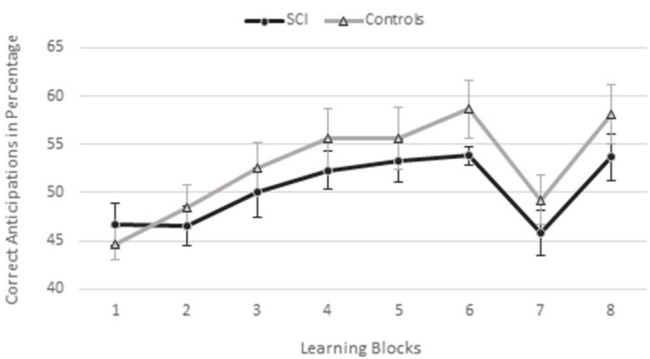

**Fig 3. Ocular activated serial reaction time task anticipation scores (mean and SEM) in the spinal cord injury (SCI) and control groups.**

the main effect of Group, F(1, 11) = 3.80, $p$ = .078, $\eta_p^2$ = .26, nor the Group x Block interaction, F(1, 11) = 0.15, $p$ = .70, $\eta_p^2$ = .01, was significant, indicating that the two groups showed similar reaction times across blocks in this stage, and similar recovery patterns.

**Anticipation score.** The results of the O-SRT task anticipation score analyses (described above) are presented in Fig 3.

*Learning.* Across groups, there was a significant increase in anticipation score over Blocks 1–6, F(5, 55) = 11.61, $p < .001$, $\eta_p^2$ = .51, while neither the main effect of Group, $F(1, 11)$ = 0.61, $p$ = .45, $\eta_p^2$ = .05, nor the Group x Block interaction, $F(5, 55)$ = 1.21, $p$ = .32, $\eta_p^2$ = .10, was significant. These results demonstrate that anticipation scores increased in both groups over the learning blocks and that both groups showed similar learning patterns.

*Interference.* There was a main effect of Block, $F(1, 11)$ = 29.64, $p < .001$, $\eta_p^2$ = .73, indicating that the anticipation scores in interference Block 7 were significantly lower than in the preceding Block 6 (i.e., interference effect), across groups. Neither the main effect of Group, $F(1, 11)$ = 3.32, $p$ = .01, $\eta_p^2$ = .23, nor the Group x Block interaction, $F(1, 11)$ = 0.05, $p$ = .83, $\eta_p^2$ = .005, reached significance, indicating that the two groups had similar interference patterns.

*Recovery.* There was a main effect of Block, $F(1, 11)$ = 30.03, $p < .001$, $\eta_p^2$ = .73, indicating higher anticipation scores in Block 8 than in Block 7 (i.e., recovery effect), across groups. Neither the main effect of Group, $F(1, 11)$ = 2.81, $p$ = .12, $\eta_p^2$ = .20, nor the Group x Block interaction, $F(1, 11)$ = 0.02, $p$ = .89, $\eta_p^2$ = .002, reached significance, indicating that the two groups showed similar anticipation scores and similar recovery patterns in this stage.

**Power analysis for O-SRT learning effects.** A post-hoc analysis of the within-subjects factor Learning (O-SRT blocks 1–6), using the effect size of $\eta_p^2$ = .51 found in the repeated measures ANOVA with anticipation score as the dependent variable, revealed a power (1-β) of 0.99. Learning effect size for reaction time was higher, at $\eta_p^2$ = .58, such that the repeated measures ANOVA for this variable would have even higher power than the anticipation score analysis.

## Awareness

The Awareness score of the SCI group did not significantly differ from that of the control group (Z = 0.63, $p$ = .53).

## Discussion

To our knowledge, the current study is the first to use an ocular activated version of the SRT task [10] to directly examine sequence learning in individuals with tetraplegia, as part of a

broader effort to characterize this function in individuals with SCI of varying types and degrees. The rationale for this examination lies in the deep reliance of post-acute SCI rehabilitation on the learning and implementation of radically new skills [23, 24]. Implicit procedural learning plays a key role in this process, making it essential to successful rehabilitation and adaptation to daily life tasks post-injury.

In accordance with clinically-observed procedural learning difficulties, previous work in our lab revealed a specific deficit in implicit sequence learning among individuals with paraplegia using a standard, manual version of the SRT task [2]. In contrast, participants with tetraplegia in the current study, like controls, exhibited gradually decreasing reaction times and increasing anticipation scores over the course of the oculomotor task, indicating intact implicit sequence learning in this group. The current work did not include a paraplegia group, precluding firm conclusions regarding SCI level-related differences in learning (see further discussion in the Study Limitations and Future Research section below). However, the differential findings of the two studies do prompt preliminary hypotheses regarding the nature of potential procedural learning deficits in SCI, and can thereby inform future research.

Differences in the extent of paralysis between SCI-related paraplegia and tetraplegia result from the height of injury, with the former involving injuries at or below the thoracic level and the latter at the cervical level [13]. Tetraplegia, considered the more severe condition, has also been associated with greater secondary structural changes in the brain [25]. As such, all things held equal, we might expect neurocognitive deficits found in individuals with paraplegia to be replicated or even more pronounced in individuals with tetraplegia. The fact that this did not occur in the current study raises the possibility that the reduced sequence learning observed in paraplegia was specific to the manual response, shedding light on potential differences between the neural systems and processes on which the two tasks rely.

Manual and ocular activated versions of the SRT task have been shown to elicit similar response patterns in healthy participants [10], suggesting that they tap a common sequence learning component. However, the conflicting results of our manual SRT (paraplegia) and O-SRT (tetraplegia) studies support the involvement of additional, modality-dependent processes. Implicit procedural learning is believed to involve both perceptual and motor components, associated with stimulus sequences and motor response sequences, respectively [26–29]. Though the relative contributions of these components to learning are still under debate, there is work showing that they are mediated by different neural systems. For example, a fMRI study by Rose et al. [30] showed activation in the hippocampus that was exclusively related to learning of a visual SRT sequence and not to motor sequence learning which, in contrast, recruited the basal ganglia and motor cortex regions. Assuming that the stimulus-dependent (perceptual) component was intact, reduced learning resulting from SCI-related deficits in the response-dependent (motor) component of the manual SRT task would not be expressed in the ocular activated version. We can speculate that the motor component specific to the manual SRT task explains the discrepancy between our findings in paraplegia and in tetraplegia. However, a direct comparison between the two populations (showing reduced learning in paraplegia alongside intact learning in tetraplegia) would be required to draw this conclusion. Further support for this explanation could be demonstrated by comparing the manual and ocular tasks in individuals with paraplegia; if a specific deficit in the motor component of sequence learning is associated with SCI, we would expect them to show deficits in the manual, but not the ocular, SRT task.

An alternative explanation for the discrepancy between our current and previous results involves differences in time since injury. Participants with SCI in the current study were in the chronic phase, with an average time since injury of over three years. Thus, they had presumably undergone substantial recovery and compensatory processes. In contrast, all but one

participant in the paraplegia study [2] were tested within six months of injury. It is possible that specific sequence learning deficits are associated with the early stages of post-SCI recovery, but diminish with time. In this case, deficits would not be evident in the current sample of chronic patients, even if they had been present closer to the time of injury. This possibility can be clarified through further research examining O-SRT performance in individuals with tetraplegia within six months of being injured.

Beyond contributing to our understanding of sequence learning processes, our findings support the viability of the O-SRT task as a measure of implicit procedural learning in populations that are unable to perform the manual version. The capacity to assess this skill in individuals with tetraplegia and related conditions is of clear theoretical and clinical value, as it can potentially guide interventions to improve rehabilitation efficacy and daily life functioning, while enabling research on the neural mechanisms of procedural learning in general.

Evaluating learning, memory, and other cognitive functions is crucial to successful rehabilitation in individuals with impaired motor functions [23, 31], for whom there are specialized standards of practice for assessment [32]. Impairment or complete loss of motor function in the upper extremities is particularly challenging in this context, due to limitations on commonly-used response modalities such as picking up objects, manually manipulating stimuli, striking a keyboard, pointing, or using a writing utensil [33]. As such, cognitive functioning following high spinal cord injury is often appraised using tasks that rely on or have been adapted for use with verbal responses. When verbal functioning is impaired by the injury as well, or when the verbal response can confound results, the ability to assess cognition becomes even more limited.

Going beyond the evaluation of procedural learning, the current findings suggest that the pool of available assessment tools for individuals with tetraplegia can be expanded by adapting standard tasks to employ oculomotor responses. This methodology stands to diversify the functions that can be tested in individuals with high SCI and similar populations, while offering solutions for those who have impaired verbal as well as motor functions. From a clinical standpoint, the potential applicability of oculomotor responses in assessing learning following SCI is strengthened by recent developments in the utilization of eye movements for communication and locomotion [34, 35]. Like many cognitive functions, learning is often influenced by and dependent on context and modality. As eye movements are increasingly harnessed to enable the performance of daily functions, use of the ocular modality to assess cognition can improve ecological validity and, in turn, the accuracy of assessment.

## Study limitations and future research

The reported findings should be considered in the context of a number of limitations, which characterize much of the clinical research on individuals with SCI. Though statistically adequate for revealing learning effects, the study sample was relatively small, due in part to the exclusion of participants with concomitant brain injury and potentially confounding premorbid conditions. It was also subject to potential confounding factors such as long-term hospitalization and medication-use. In future research, some of this variability may be limited by increasing sample sizes and including a control group of individuals who sustained traumatic injuries not involving the brain or spinal cord. Furthermore, as noted above, clear cut conclusions regarding the effects of SCI level on procedural learning require a direct comparison between paraplegia and tetraplegia groups using the ocular task. This type of study could also clarify the motor versus cognitive components of potential SCI-related learning deficits, as could the comparison of manual and ocular tasks in individuals with paraplegia. To address the role of time since injury and potential confounding effects, these studies should be

conducted within six months of injury, or include between-group comparisons based on this variable. Larger sample sizes would also allow us to examine correlations between various learning measures and time since injury. Behavioral studies should also be complemented with imaging studies, which may serve to explain and predict interpersonal and between-group differences in procedural learning following SCI and to reveal the neural processes involved in sequence learning tasks with different response modalities.

## Conclusions

Individuals with tetraplegia demonstrated unimpaired sequence learning on an ocular activated serial reaction time task, with performance patterns similar to those found in healthy populations using standard motor versions of the task. This suggests that previously reported implicit sequence learning deficits in spinal cord injury directly involved motor functioning rather than cognitive aspects of the task. The ocular activated sequence learning task could be a valid alternative for assessing implicit sequence learning in populations that cannot perform spinal-cord dependent motor tasks, with important implications for post-SCI rehabilitation and adjustment.

## Supporting information

**S1 Dataset. Raw data from serial reaction time (SRT) task, depression, and additional cognitive measures, for control and spinal cord injury (SCI) groups.**
(XLSX)

## Acknowledgments

The research presented in this paper was conducted within the context of Michal Shaham's master's degree at Ariel University.

## Author Contributions

**Conceptualization:** Ayala Bloch, Michal Shaham, Eli Vakil, Gabi Zeilig.

**Data curation:** Ayala Bloch, Michal Shaham, Eli Vakil, Simone Schwizer Ashkenazi.

**Formal analysis:** Ayala Bloch, Michal Shaham, Eli Vakil, Simone Schwizer Ashkenazi.

**Investigation:** Ayala Bloch, Michal Shaham, Eli Vakil, Gabi Zeilig.

**Methodology:** Ayala Bloch, Michal Shaham, Eli Vakil, Gabi Zeilig.

**Project administration:** Michal Shaham.

**Resources:** Ayala Bloch, Eli Vakil, Gabi Zeilig.

**Supervision:** Ayala Bloch.

**Writing – original draft:** Ayala Bloch, Michal Shaham.

**Writing – review & editing:** Ayala Bloch, Michal Shaham, Eli Vakil, Simone Schwizer Ashkenazi, Gabi Zeilig.

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
