## [Decision Letter · Decision Letter 0]

17 Jan 2020

PONE-D-19-35143

Examining Implicit Procedural Learning in Tetraplegia Using an Oculomotor Serial Reaction Time Task

PLOS ONE

Dear Dr Ayala Bloch,

Thank you for submitting your manuscript to PLOS ONE. After careful consideration, we feel that it has merit but does not fully meet PLOS ONE’s publication criteria as it currently stands. Therefore, we invite you to submit a revised version of the manuscript that addresses the points raised during the review process.

We would appreciate receiving your revised manuscript by 45 days. To enhance the reproducibility of your results, we recommend that if applicable you deposit your laboratory protocols in protocols.io, where a protocol can be assigned its own identifier (DOI) such that it can be cited independently in the future. For instructions see: http://journals.plos.org/plosone/s/submission-guidelines#loc-laboratory-protocols

We look forward to receiving your revised manuscript.

Kind regards,

Mariella Pazzaglia

Academic Editor

PLOS ONE

Journal Requirements:

2. We note that you have indicated that you obtained written consent. Please clarify how consent was recorded for paraplegic participants.

3. Please ensure that you refer to Figure 2 in your text as, if accepted, production will need this reference to link the reader to the figure.

Reviewers' comments:

Reviewer's Responses to Questions

**Comments to the Author**

1. Is the manuscript technically sound, and do the data support the conclusions?

Reviewer #1: Yes

Reviewer #2: Yes

2. Has the statistical analysis been performed appropriately and rigorously? 

Reviewer #1: Yes

Reviewer #2: Yes

3. Have the authors made all data underlying the findings in their manuscript fully available?

Reviewer #1: Yes

Reviewer #2: Yes

4. Is the manuscript presented in an intelligible fashion and written in standard English?

Reviewer #1: Yes

Reviewer #2: Yes

5. Review Comments to the Author

Reviewer #1: In their work, the authors explore the effects of chronic tetraplegia on implicit procedural learning using an oculomotor version of the serial reaction time (SRT) task. An experimental group of 12 tetraplegic patients with a spinal cord injury (SCI) and a paired group of healthy participants are asked to complete the task using an eye-tracking device. The main finding of this work is that tetraplegic patients using the oculomotor version of the SRT task do not show alterations of implicit learning pattern. This goes in the opposite direction of (their) previous research experience. Moreover, the authors demonstrate the usefulness of the oculomotor version of the SRT in patients with reduced mobility. Demonstrating that procedural learning is not impaired underlines the importance of proper rehabilitation procedures.

The article is well written and easy to understand. The findings are relevant and adequately supported.

Comments:

-Several times, throughout the paper, the authors cite their previous works and offer interpretations for the differing results. Even if the inferences are supported by literature, something is missing: a second control group with SCI patients with chronic paraplegia. Although this is a demanding request, the suggestion is to close the gap that is missing from the proposed statements by introducing a new control group made up of paraplegic patients (with a chronic SCI). Proposed inferences from previous works may be theoretically reasonable, but experimentally there is no direct comparison between paraplegic and tetraplegic patients. Moreover, different time since lesion may differently influence performance and manual and oculomotor versions may have hidden differences. These two points are highlighted in the discussion but should have a broader impact on the manuscript.

-In any case, given what said above, a differently structured discussion may be considered useful, highlighting the limits of this work in a special section and eventually a brief conclusion section.

- Probably it is not relevant but had tetraplegic participants previous experience in using eye tracking-based devices? If yes, are there possible implications?

Reviewer #2: The manuscript describes a technically sound piece of scientific research that will advance our knowledge of cognitive impairment in SCI. It is a well conducted work, meaningful, with a clear explanation of what the impact of this knowledge on clinical practice is. Experiments have been conducted rigorously, and their explanation allows to replicate them, with appropriate controls. The conclusions are drawn appropriately based on the data presented. The point on time since injury differences between this study and a previous one is of particular relevance. It would be worth considering correlating the performance at the task with clinical variables such as time since onset to see if there is any relationship, as done in previous studies on SCI. If data allows.

A couple of points though are worth mentioning that I believe if addressed would improve the manuscript impact:

- As power analysis was not conducted, though, we cannot really establish if the sample is large enough to consider the findings reliable. I wonder if the Authors are able to include a power analysis in their manuscript.

- The work would have been much more robust had the Authors tested also paraplegic patients. The conclusions they can draw, despite being correct, are very limited, given this has not been done. As the authors seem to have access to these patients I wonder why this sample has not been included, and if it is possible to include them.

Other comments

- Abstract very clear

- Line 48 (introduction): provide an example of these effects

- Line 64 (introduction): by other variables such as? I would assume maybe cognitive ones? Worth saying a couple of them.

- Very good introduction, focused, clearly making a point also on the validity of the task used. It would be worth though recognizing that the use of eye movement in tetraplegic patients has been adopted before, as a good alternative to tasks requiring a verbal response or a hand response (see Brain and Cognition, 73, 189–193. doi:10.1016/j.bandc- .2010.05.001 and J Neuropsychol. 2014 Sep;8(2):199-215. doi: 10.1111/jnp.12020).

- The Authors present the variable marital status but I am not clear how this is relevant for this study. I see they use it to match controls, but it is not necessary to match participants in terms of marital status for this specific task. Or the Authors could explain how marital status is going to make a difference in terms of matching samples?

- Verbal and performance IQ selected to measure possible confounding factors. Why are these and no other measures (such as visual search through eye movement) explored? One could infer that general intelligence is what is aimed at here, to ensure it is not a cognitive impairment, however also other basic functions could explain the results? It seems clearer from the digit span and matrix explanation at line 191, but it is not clear why verbal performance matters. Is it related to instructions? In an effort to allow replicability I would explain this choice better.

- Does the order matter? Are the last tests administered after to avoid tiredness?

- Check degrees of freedom in analyses: I am not entirely sure the ones indicated for the interactions are correct.

- The data are available. However, no legend is provided with the file making it difficult to understand the meaning of the labels. Some explanation is provided within the labels, such as for gender. However, some others are unclear: for instance, “Matching ID (same numbers matched)”, what does it mean? A full legend in a second sheet would be better.

6. PLOS authors have the option to publish the peer review history of their article (what does this mean?). If published, this will include your full peer review and any attached files.

Reviewer #1: No

Reviewer #2: No

---

## [Author Response · Author response to Decision Letter 0]

28 Feb 2020

February 26, 2020

Editor-in-Chief Joerg Heber

PLOS ONE

Dear Dr. Heber,

Thank you for considering our manuscript, " Examining implicit procedural learning in tetraplegia using an oculomotor serial reaction time task," for publication in PLOS ONE. We have read and considered the reviewers' comments carefully and are grateful for their constructive suggestions and for the time and effort they invested in helping us to improve this submission. 

Following revisions in accordance with the reviewers' suggestions, we are resubmitting our manuscript (with and without tracked changes) alongside a detailed response addressing each comment (below). We hope that you will find the revised paper suitable for publication in PLOS ONE.

Sincerely,

Ayala Bloch

Corresponding Author

Journal Requirements:

Response: Please note that we were unable to open the links above (received error messages that the pages could not be found). The manuscript is now formatted in accordance with the following links and additional guidelines found on the PLOS ONE website: https://journals.plos.org/plosone/s/file?id=80c1/PLOSOne_formatting_sample_main_body.pdf;
https://journals.plos.org/plosone/s/file?id=ba62/PLOSOne_formatting_sample_title_authors_affiliations.pdf

2. We note that you have indicated that you obtained written consent. Please clarify how consent was recorded for paraplegic participants.

Response: When physically able, participants signed a written informed consent form; otherwise, fingerprints were used to indicate informed consent. This is indicated in the Methods section of the revised manuscript (page 9, lines 145-6).

3. Please ensure that you refer to Figure 2 in your text as, if accepted, production will need this reference to link the reader to the figure.

Response: The reference to Figure 2 was accidentally omitted, and Figure 3 was referenced twice. This has been corrected such that the revised manuscript contains a reference to Figure 2 (page 14, line 253).

Reviewers' comments:

Reviewer's Responses to Questions

Comments to the Author:

1. Is the manuscript technically sound, and do the data support the conclusions?

Reviewer #1: Yes

Reviewer #2: Yes

2. Has the statistical analysis been performed appropriately and rigorously?

Reviewer #1: Yes

Reviewer #2: Yes

3. Have the authors made all data underlying the findings in their manuscript fully available?

Reviewer #1: Yes

Reviewer #2: Yes

Reviewer #1: In their work, the authors explore the effects of chronic tetraplegia on implicit procedural learning using an oculomotor version of the serial reaction time (SRT) task. An experimental group of 12 tetraplegic patients with a spinal cord injury (SCI) and a paired group of healthy participants are asked to complete the task using an eye-tracking device. The main finding of this work is that tetraplegic patients using the oculomotor version of the SRT task do not show alterations of implicit learning pattern. This goes in the opposite direction of (their) previous research experience. Moreover, the authors demonstrate the usefulness of the oculomotor version of the SRT in patients with reduced mobility. Demonstrating that procedural learning is not impaired underlines the importance of proper rehabilitation procedures.

The article is well written and easy to understand. The findings are relevant and adequately supported.

Comments:

-Several times, throughout the paper, the authors cite their previous works and offer interpretations for the differing results. Even if the inferences are supported by literature, something is missing: a second control group with SCI patients with chronic paraplegia. Although this is a demanding request, the suggestion is to close the gap that is missing from the proposed statements by introducing a new control group made up of paraplegic patients (with a chronic SCI). Proposed inferences from previous works may be theoretically reasonable, but experimentally there is no direct comparison between paraplegic and tetraplegic patients. Moreover, different time since lesion may differently influence performance and manual and oculomotor versions may have hidden differences. These two points are highlighted in the discussion but should have a broader impact on the manuscript.

-In any case, given what said above, a differently structured discussion may be considered useful, highlighting the limits of this work in a special section and eventually a brief conclusion section.

Response: We thank the reviewer for this comment and recognize its validity and significance. As noted, we did originally highlight the lack of direct comparison and the potential effect of time since legion in the Discussion, but agree that there is room for further emphasis. 

The addition of a new control group would be difficult at this point: during original data collection, our stringent exclusion criteria resulted in an extremely long patient recruitment process and we believe adding another group could delay publication of these findings by over a year. We strongly believe, however, that the findings are of value in their current form, particularly to encourage and inform further research (including direct comparisons between tetraplegia and paraplegia). As such, we have revised the Discussion in accordance with the reviewer’s recommendations, as follows: 

• We added the following section toward the beginning of the Discussion (page 17, lines 319-23): “The current work did not include a paraplegia group, precluding firm conclusions regarding SCI level-related differences in learning (see further discussion in the Study Limitations and Future Research section below). However, the differential findings of the two studies do prompt preliminary hypotheses regarding the nature of potential procedural learning deficits in SCI, and can thereby inform future research.”

• We added a Study Limitations and Future Research heading (page 20, line 390), and added the following text to the existing paragraph (pages 20-1, lines 398-405: “Furthermore, as noted above, clear cut conclusions regarding the effects of SCI level on procedural learning require a direct comparison between paraplegia and tetraplegia groups using the ocular task. This type of study could also clarify the motor versus cognitive components of potential SCI-related learning deficits, as could the comparison of manual and ocular tasks in individuals with paraplegia. To address the role of time since injury and potential confounding effects, these studies should be conducted within six months of injury, or include between-group comparisons based on this variable. Larger sample sizes would also allow us to examine correlations between various learning measures and time since injury.”

• We added a brief Conclusions section (page 21, lines 410-8).

- Probably it is not relevant but had tetraplegic participants previous experience in using eye tracking-based devices? If yes, are there possible implications?

Response: This is a good question, to which we unfortunately do not have the answer. It is worth noting, however, that we have no reason to expect that general practice/experience in using eye tracking-based devices would affect sequence learning specifically. If anything, between-group differences in practice would be more likely to affect baseline speeds and anticipation, resulting in main effects of Group – but we did not find such effects for any of the SRT measures. 

Reviewer #2: The manuscript describes a technically sound piece of scientific research that will advance our knowledge of cognitive impairment in SCI. It is a well conducted work, meaningful, with a clear explanation of what the impact of this knowledge on clinical practice is. Experiments have been conducted rigorously, and their explanation allows to replicate them, with appropriate controls. The conclusions are drawn appropriately based on the data presented. The point on time since injury differences between this study and a previous one is of particular relevance. It would be worth considering correlating the performance at the task with clinical variables such as time since onset to see if there is any relationship, as done in previous studies on SCI. If data allows.

Response: We thank the reviewer for the positive feedback and for the valid comment on time since injury. We did conduct simple correlation analyses to examine potential associations between time since injury and measures of learning, and did not find any significant correlations. However, we chose not to report these results, as our sample size was too low to give the correlation analyses sufficient power. Still, in accordance with this comment, we chose to place more emphasis on this issue in the (newly added) Study Limitations and Future Research section, which now includes the following text (pages 20-1, lines 402-5): “To address the role of time since injury and potential confounding effects, these studies should be conducted within six months of injury, or include between-group comparisons based on this variable. Larger sample sizes would also allow us to examine correlations between various learning measures and time since injury.”

A couple of points though are worth mentioning that I believe if addressed would improve the manuscript impact:

- As power analysis was not conducted, though, we cannot really establish if the sample is large enough to consider the findings reliable. I wonder if the Authors are able to include a power analysis in their manuscript.

Response: When collecting data for this study, we did not originally conduct a power analysis, but rather based our sample sizes on a significant number of spinal cord injury studies with similar or smaller participant numbers. In response to this comment, we conducted a post-hoc analysis of the within-subjects factor Learning (SRT blocks 1-6), using the effect size of ηp² = .51 found in the RM ANOVA with anticipation score as the dependent variable, which revealed a power (1-β) of 0.99. Note that learning effect size for reaction time was higher, at ηp² = .58, such that the RM ANOVA for this variable would have even higher power than the anticipation score analysis. We have included this information in the revision (page 16, lines 296-301). 

Regardless of the adequate power for the learning analyses, it is clear that larger group sizes would increase the reliability of the study, and that future studies should aim to include more participants. This is more clearly noted in the revision, in a new Study Limitations and Future Research section (page 20, lines 392-8).

- The work would have been much more robust had the Authors tested also paraplegic patients. The conclusions they can draw, despite being correct, are very limited, given this has not been done. As the authors seem to have access to these patients I wonder why this sample has not been included, and if it is possible to include them.

Response: We thank the reviewer for this comment and recognize its validity and significance. We did originally note the limitations associated with the lack of direct paraplegia-tetraplegia comparison (which was beyond the scope of original data collection for technical reasons) in the Discussion, but understand that there is room for further emphasis. 

The addition of a new control group would be difficult at this point: during original data collection, our stringent exclusion criteria resulted in an extremely long patient recruitment process and we believe adding another group could delay publication of these findings by over a year. We strongly believe, however, that the findings are of value in their current form, particularly to encourage and inform further research (including direct comparisons between tetraplegia and paraplegia). As such, we have revised the Discussion as follows: 

• We added the following section toward the beginning of the Discussion (page 17, lines 319-323): “The current work did not include a paraplegia group, precluding firm conclusions regarding SCI level-related differences in learning (see further discussion in the Study Limitations and Future Research section below). However, the differential findings of the two studies do prompt preliminary hypotheses regarding the nature of potential procedural learning deficits in SCI, and can thereby inform future research.”

• We added a Study Limitations and Future Research heading (page 20, line 390), and added the following text to the existing paragraph (pages 20-1, lines 398-401): “Furthermore, as noted above, clear cut conclusions regarding the effects of SCI level on procedural learning require a direct comparison between paraplegia and tetraplegia groups using the ocular task. This type of study could also clarify the motor versus cognitive components of potential SCI-related learning deficits, …”

Other comments

- Abstract very clear

- Line 48 (introduction): provide an example of these effects

Response: To address this comment, the following text has been added to the Introduction (page 3, line 52-5): “This is because new skills inevitably determine the ability to effectively and independently conduct daily functions following SCI, such as communicating using eye or head motions, or proficiently activating a wheelchair using limbs with newly limited motion.”

- Line 64 (introduction): by other variables such as? I would assume maybe cognitive ones? Worth saying a couple of them.

Response: The other variables examined in Bloch et al. (2016) included mood, intelligence, and verbal and visuospatial memory. This is indicated in the revised manuscript (page 4, line 72).

- Very good introduction, focused, clearly making a point also on the validity of the task used. It would be worth though recognizing that the use of eye movement in tetraplegic patients has been adopted before, as a good alternative to tasks requiring a verbal response or a hand response (see Brain and Cognition, 73, 189–193. doi:10.1016/j.bandc- .2010.05.001 and J Neuropsychol. 2014 Sep;8(2):199-215. doi: 10.1111/jnp.12020).

Response: Thank you for the positive feedback regarding the Introduction and for the suggestion that we recognize the use of eye movements to adapt other tasks for individuals with tetraplegia. The proposed papers are noted and referenced in the revised manuscript (page 4, line 85-7).

- The Authors present the variable marital status but I am not clear how this is relevant for this study. I see they use it to match controls, but it is not necessary to match participants in terms of marital status for this specific task. Or the Authors could explain how marital status is going to make a difference in terms of matching samples?

Response: Thank you for drawing our attention to this mistake in the original manuscript. We collected marital status as a demographic variable, but did not use it for matching, which was based only on age, sex, and education. Marital status is indeed irrelevant to the current study and has therefore been omitted from the revised manuscript.

- Verbal and performance IQ selected to measure possible confounding factors. Why are these and no other measures (such as visual search through eye movement) explored? One could infer that general intelligence is what is aimed at here, to ensure it is not a cognitive impairment, however also other basic functions could explain the results? It seems clearer from the digit span and matrix explanation at line 191, but it is not clear why verbal performance matters. Is it related to instructions? In an effort to allow replicability I would explain this choice better.

Response: Verbal and performance IQ measures (Vocabulary and Matrix Reasoning) were included in the study first and foremost as a basis for inclusion (at least low average) and to ensure that instructions were understood. All of the participants met this criterion, so we did not originally report it, but agree that this can cause confusion and have therefore included it now (page 6, line 116-117). Also, had there been any between-group effects in the learning measures, we would have liked to ensure that they were not explained by between-group differences in IQ or in Digit Span, as a measure of explicit memory/learning.

- Does the order matter? Are the last tests administered after to avoid tiredness?

Response: The Awareness measure had to follow the O-SRT, as it addresses explicit knowledge of the sequences presented. Beyond this, there was no particular rationale for the order, but it was followed consistently.

- Check degrees of freedom in analyses: I am not entirely sure the ones indicated for the interactions are correct.

Response: After additional examination, we confirm that the degrees of freedom for the two-way matched-subject repeated measures ANOVAs are correct.

- The data are available. However, no legend is provided with the file making it difficult to understand the meaning of the labels. Some explanation is provided within the labels, such as for gender. However, some others are unclear: for instance, “Matching ID (same numbers matched)”, what does it mean? A full legend in a second sheet would be better.

Response: In accordance with this comment, a revised data set has been made available, with a full legend in a second sheet.

---

## [Decision Letter · Decision Letter 1]

8 Apr 2020

Examining Implicit Procedural Learning in Tetraplegia Using an Oculomotor Serial Reaction Time Task

PONE-D-19-35143R1

Dear Dr. Bloch,

We are pleased to inform you that your manuscript has been judged scientifically suitable for publication and will be formally accepted for publication once it complies with all outstanding technical requirements.

With kind regards,

Mariella Pazzaglia

Academic Editor

PLOS ONE

Additional Editor Comments (optional):

Reviewers' comments:

Reviewer's Responses to Questions

**Comments to the Author**

1. If the authors have adequately addressed your comments raised in a previous round of review and you feel that this manuscript is now acceptable for publication, you may indicate that here to bypass the “Comments to the Author” section, enter your conflict of interest statement in the “Confidential to Editor” section, and submit your "Accept" recommendation.

Reviewer #1: All comments have been addressed

Reviewer #2: All comments have been addressed

2. Is the manuscript technically sound, and do the data support the conclusions?

Reviewer #1: Yes

Reviewer #2: Yes

3. Has the statistical analysis been performed appropriately and rigorously? 

Reviewer #1: Yes

Reviewer #2: Yes

4. Have the authors made all data underlying the findings in their manuscript fully available?

Reviewer #1: Yes

Reviewer #2: Yes

5. Is the manuscript presented in an intelligible fashion and written in standard English?

Reviewer #1: Yes

Reviewer #2: Yes

6. Review Comments to the Author

Reviewer #1: (No Response)

Reviewer #2: Thanks for addressing my comments. The manuscript is now ready for publication in my view. I am sure this will become a reference point in the study of sci from a cognitive point of view.

7. PLOS authors have the option to publish the peer review history of their article (what does this mean?). If published, this will include your full peer review and any attached files.

Reviewer #1: No

Reviewer #2: No

---

## [Editor Report · Acceptance letter]

13 Apr 2020

PONE-D-19-35143R1 

Examining Implicit Procedural Learning in Tetraplegia Using an Oculomotor Serial Reaction Time Task 

Dear Dr. Bloch:

I am pleased to inform you that your manuscript has been deemed suitable for publication in PLOS ONE. Congratulations! Your manuscript is now with our production department. 

With kind regards,

on behalf of

Dr. Mariella Pazzaglia 

Academic Editor

PLOS ONE